# Basin Entropy and Shearless Barrier Breakup in Open Non-Twist Hamiltonian Systems

**DOI:** 10.3390/e25081142

**Published:** 2023-07-30

**Authors:** Leonardo C. Souza, Amanda C. Mathias, Pedro Haerter, Ricardo L. Viana

**Affiliations:** Departamento de Física, Universidade Federal do Paraná, Curitiba 81531-990, PR, Brazil; leonardo.ucoz@gmail.com (L.C.S.); amaandafisica@gmail.com (A.C.M.); pedrohaerter.095@gmail.com (P.H.)

**Keywords:** basin entropy, shearless barriers, non-twist maps, open Hamiltonian systems

## Abstract

We consider open non-twist Hamiltonian systems represented by an area-preserving two-dimensional map describing incompressible planar flows in the reference frame of a propagating wave, and possessing exits through which map orbits can escape. The corresponding escape basins have a fractal nature that can be revealed by the so-called basin entropy, a novel concept developed to quantify final-state uncertainty in dynamical systems. Since the map considered violates locally the twist condition, there is a shearless barrier that prevents global chaotic transport. In this paper, we show that it is possible to determine the shearless barrier breakup by considering the variation in the escape basin entropy with a tunable parameter.

## 1. Introduction

The study of the non-integrable Hamiltonian system is one of the main disciplines in the field of nonlinear dynamics [1]. A large number of Hamiltonian systems of physical interest can be analytically and numerically investigated through the use of area-preserving mappings in a Poincaré surface of section of the phase space. In particular, those systems satisfying the so-called *twist property* for which, loosely speaking, there are no orbits with the same frequency in terms of the corresponding angle variables in phase space [2], have been most extensively studied. One outstanding example of area-preserving mappings satisfying the twist property is the Chirikov–Taylor map [3].

One major advantage of working with area-preserving twist maps is that many powerful results of Hamiltonian theory such as KAM and Poincaré–Birkhoff theorems and Aubry–Mather theory are valid provided the system obeys the twist property [2]. However, in various Hamiltonian systems of physical interest, chiefly fluids and plasmas, the twist property fails to be satisfied, which has motivated the study of area-preserving non-twist maps [4]. One paradigmatic example of the latter category is the standard non-twist map introduced by del Castillo-Negrete and Morrison, for which the twist condition is locally violated [5].

The non-twist character of such maps has a profound influence on their dynamical properties [6]. For example, due to the non-monotonic character of the frequency profile, there appear twin island chains failing to exhibit the well-known island overlapping. Instead, the islands suffer a kind of reconnection process and produce robust shearless transport barriers that modify the transport properties displayed by non-twist maps [7].

Among the wide variety of physically relevant systems described by area-preserving non-twist maps, we mention the magnetic field line structure in tokamaks and stellarators [8,9,10], planetary orbits [11,12], stellar pulsations [13], atomic physics [14,15], condensed matter [16] and sheared geostrophic flows [5,17]. In all these non-twist systems, the existence of shearless transport barriers represent a local obstruction to chaotic diffusion of phase space trajectories. One of the relevant problems involving non-twist systems is how to characterize numerically the destruction of those shearless barriers [18]. This problem has been investigated in considerable detail for the standard non-twist map, thanks to a special property that is the existence of the so-called indicator points [19].

For general area-preserving non-twist maps, however, the absence of such indicator points makes the numerical task of determining the breakup of the shearless barrier a difficult one. Some methods have been proposed for this task. In the present paper, we propose an alternative method to determine the shearless curve breakup by using a definition of entropy applied to basins of behavior [20,21]. The latter is an extension, for general dynamical systems, of the usual basin of attraction concept. Since Hamiltonian systems do not possess attractors, we can define an analogous behavior by opening its domain and considering the escape of trajectories. In this sense, the basin of escape is the set of initial conditions (in the Poincaré surface of section) generating trajectories that escape through that exit.

Due to the underlying structure of the dynamics in a chaotic orbit of a non-integrable Hamiltonian system, the structure of escape basins is highly fractal [22]. The basin entropy quantifies the uncertainty related to the fractality of the escape basins and of their common boundary, and has been used in many Hamiltonian systems with this purpose. In the present work, however, we used the basin entropy specifically to determine for which value of the perturbation parameter (measuring, so as to speak, the strength of the system non-integrability) the shearless barrier suffers a breakup.

This paper is structured as follows: in Section 2, we introduce the specific area-preserving non-twist map used. In Section 3, we exhibit the escape basin structure and its characterization using an uncertainty exponent. Section 4 introduces the concept of basin entropy for the escape basins of an open Hamiltonian system. In Section 5, we show how to use this concept to characterize the shearless barrier breakup. Our Conclusions are left to the final section.

## 2. Area-Preserving Non-Twist Maps

Let us consider a Hamiltonian system with *N* degrees of freedom, characterized by canonical pairs $\left(p_{i},q_{i}\right)$, i=1,2,…N. This is an integrable system if one can obtain a symplectic transformation to action-angle coordinates
(1)(I,θ)→(q(I,θ),p(I,θ)),
where $I=\{I_{1},I_{2},\ldots I_{N}\}\in B\subset N$ (*B* is an open set) and $\theta =\{\left(\phi_{1},\phi_{2},\ldots \phi_{N}\right)\;\mathrm{mod}\;2\pi \}$, such that $\phi_{i}$ parameterize the motion on a *N*-dimensional torus. In terms of these action-angle coordinates, the Hamiltonian becomes
(2)H(q(I,θ),p(I,θ))=K(I),
and the corresponding Hamilton’s equations are
(3)$$\begin{array}{cc} \frac{dI}{dt} & =-\frac{\partial K(I)}{\partial \theta }=0, \end{array}$$
(4)$$\begin{array}{cc} \;\frac{d\theta }{dt} & =\frac{\partial K(I)}{\partial I}=\omega \left(I\right), \end{array}$$
where $\omega \in \{\left(\omega_{1},\omega_{2},\ldots \omega_{N}\right)\}$ are the frequencies corresponding to each irreducible circuit on the *N*-torus.

Provided that the energy surfaces in the phase space are closed and bounded, a one-degree of freedom system with a time-independent Hamiltonian is integrable, so that, the simplest non-integrable systems have N=2. In addition, let us consider that the non-integrability comes from a weak perturbation of an integrable system, in the standard form
(5)$$H\left(I_{1},I_{2};\theta_{1},\theta_{2}\right)=H_{0}\left(I_{1},I_{2}\right)+\varepsilon H_{1}\left(I_{1},I_{2};\theta_{1},\theta_{2}\right),$$
where ε≪1 for a quasi-integrable system. The integrable system is characterized by two frequencies, $\omega_{1}\left(I_{1},I_{2}\right)$ and $\omega_{2}\left(I_{1},I_{2}\right)$. The Hamiltonian $H_{0}$ is said to satisfy the so-called *twist condition* if
(6)$$\frac{\partial \omega_{i}}{\partial I_{j}}=\frac{\partial H_{0}}{\partial I_{j}\partial I_{i}}\neq 0,$$
i.e., the integrable system does not have two phase-space trajectories with the same frequency. Conversely, if this condition fails to be satisfied at any value of the action, the system is said to be *non-twist*.

Many Hamiltonian systems of physical interest satisfy the twist property. Moreover, for this class of systems many powerful results of Hamiltonian theory are valid, such as Kolmogorov–Arnold-Moser theorem, Poincaré–Birkhoff theorem, Aubry–Mather theory, and so on. On the other hand, there are non-twist systems of interest, mainly in hydrodynamics and plasma physics, as commented on in the Introduction. For non-twist systems, there are different dynamical properties that have been investigated in recent years [23,24].

The energy H=E is a constant of the motion inasmuch that the Hamiltonian does not depend explicitly on time. Hence, the motion, which occurs in a 4-dimensional phase space, actually is limited to a 3-dimensional energy surface $H=H(I_{1},E;\theta_{1},\theta_{2})$. Moreover, considering a Poincaré surface of section $\theta_{2}=const.$mod2π, we can reduce the continuous-time flow generated by solving (Equation 3) and (Equation 4) to a discrete-time mapping in the plane $I_{1}\times \theta_{1}$, with the general form
(7)$$\begin{array}{cc} I_{n+1} & =I_{n}+\varepsilon h\left(\theta_{n},I_{n+1}\right), \end{array}$$
(8)$$\begin{array}{cc} \;\theta_{n+1} & =\theta_{n}+f\left(I_{n+1}\right)+\varepsilon g\left(\theta_{n},I_{n+1}\right)\;\;\mathrm{mod}\;2\pi , \end{array}$$
where f(I) is the so-called winding number, and *h* and *g* represent the effects of the perturbation term in the Poincaré map. The twist condition (Equation 6) reads [25]
(9)$$\frac{d\theta_{n+1}}{dI_{n+1}}\neq 0.$$

The Poincaré map preserves the symplectic area in the surface of section if
(10)$$\frac{\partial g(\theta_{n},I_{n+1})}{\partial \theta_{n}}+\frac{\partial h(\theta_{n},I_{n+1})}{\partial I_{n+1}}=0.$$ A further simplification consists in choosing $g(\theta_{n},I_{n+1})=0$ and $h\left(\theta_{n},I_{n+1}\right)=\sin\theta_{n}$, a choice that fulfills the symplectic condition (Equation 10). In this case, the twist condition (Equation 9) reduces to df/dI≠0, i.e., the winding number profile should be monotonic over the range of the action variable.

An example of an area-preserving non-twist map, where $f\left(I\right)=k\left(I^{2}-1\right)$, was introduced by Weiss [26,27] in the context of advection of passive scalars (see Appendix A)
(11)$$\begin{array}{cc} I_{n+1} & =I_{n}-k\sin\left(\theta_{n}\right) \end{array}$$
(12)$$\begin{array}{cc} \;\theta_{n+1} & =\theta_{n}+k\left(I_{n+1}^{2}-1\right)\;\;\mathrm{mod}\;2\pi , \end{array}$$
where *k* is a parameter representing the non-integrable perturbation. Since df/dI=2kI, the twist condition is not satisfied at I=0. Indeed, non-twist maps usually have non-monotonic winding number profiles. A shorthand notation for this map is x↦M(x), where x=(I,θ) and M are given by (Equation 11) and (Equation 12). Since this is a Hamiltonian system, there exists an inverse map $M^{-1}$.

In the k→0 limit, we have an identity map (I↦I,θ↦θ). For relatively small values of *k*, the system becomes non-integrable and one can observe quasiperiodic orbits spanning the entire interval [−π,π] and invariant curves inside islands centered at a periodic orbit of the map (Equation 11) and (Equation 12). This is a consequence of the non-monotonicity of the winding number profile, i.e., there will be two orbits with the same winding number (a phenomenon also known as *degeneracy*, and which appears only for non-twist systems) [28]. For a larger value of the perturbation parameter *k*, we observe such a collision of periodic orbits, involving a reconnection of the islands’ separatrices. This is actually a global bifurcation changing the topology of the orbits as some parameter is varied through a critical value.

In the Figure 1a–d, we show phase portraits generated using the Weiss map (Equation 11) and (Equation 12) for different values of the parameter *k*. Chaotic orbits near the former islands’ separatrices that no longer exist due to the homoclinic tangle formed after reconnection can be observed. A distinguished feature of non-twist maps is the presence of a robust shearless barrier between the two islands (depicted in blue in Figure 1a). Such a shearless barrier corresponds to a local extremum of the winding number profile for the map (Equation 11) and (Equation 12). This barrier prevents global transport related to the chaotic orbits; the shearless barrier clearly separates the chaotic regions near the separatrices.

An increase of *k*, however, causes the shearless barrier breakup and the mixing of the chaotic regions associated with each island (Figure 1b). The latter, on its turn, occupies a wider fraction of the phase space as *k* is further increased (Figure 1c,d). The shearless barrier breakup occurs for a critical value of *k* between 0.50 and 0.55 but a precise determination is difficult to make only from these phase portraits. In the following, we will consider a systematic way to accomplish this task.

In order to distinguish between chaotic and non-chaotic orbits, we have computed the Lyapunov exponents for this map [29]
(13)$$\lambda_{1,2}=\lim_{n\to \infty }\frac{1}{n}\log||{\mathrm{DM}}^{n}\left(x\right)\cdot u_{1,2}\left|\right|,$$
where DM is the tangent map corresponding to Equations (Equation 11) and (Equation 12) and $u_{1,2}$ are mutually orthogonal eigendirections. Due to the area-preserving nature of the Weiss’s map, it follows that $\lambda_{1}+\lambda_{2}=0$, such that it suffices to present results for the largest Lyapunov exponent $\lambda_{1}$. A color map for the latter is shown in Figure 2 for different values of the parameter *k*, and corresponding to the same values used in the phase portraits of Figure 1.

The islands’ interior, comprising quasi-periodic closed orbits, is related to vanishing Lyapunov exponents, whereas the chaotic region near the islands’ separatrices have positive values of $\lambda_{1}$ (Figure 2a). Moreover, the existence of a shearless barrier clearly separates the local islands’ separatrices. The same pattern is observed for higher *k*, but the chaotic orbits have a considerably higher value of $\lambda_{1}$, an almost tenfold increase (Figure 2b). By the same way, the shearless barrier breakup can be observed by the Lyapunov exponent colormap (Figure 2c). The value of $\lambda_{1}$ also increases for higher *k* (Figure 2d).

## 3. Escape Basins

The Hamiltonian system given by the map Equations (Equation 11) and (Equation 12) is opened by considering that the particles can escape by one or more exits in the (x,y) phase space [30,31]. The sets of initial conditions, that reach each one of the exits, after a given number of map iterations, form their corresponding basins of escape. If the exits are placed at regions with non-chaotic orbits, their basins are relatively simple. On the other hand, if the exits are placed in chaotic orbits, their corresponding escape basins are fractal, with a fractal basin boundary. This results from the properties of the chaotic saddle, an invariant non-attracting chaotic set formed by the intersections of the unstable and stable manifolds of unstable periodic orbits embedded in the chaotic region [32].

We will consider two possible square exits of width 0.2, centered at the points (0.0,−1.1) and (π−0.1,1.0) near the islands separatrices, so that we have an escape of particles for small values of the perturbation parameter. The corresponding exits are denoted by $B_{1}$ and $B_{2}$, respectively. As a matter of fact, the absolute values of the basin entropy would change according to the exit width: smaller widths would result in slightly higher values of the basin entropy [33]. However, since we are considering in this work the relative values of the basin entropy, with respect to changes in the parameter *k*, our final results would not be modified if different exit widths would be used (provided we use the same width during the variation of *k*). For each iteration of maps (Equation 11) and (Equation 12), we make the following test: if $I_{n},\theta_{n}$ inside one of the squares we stop iterations and save the values of the initial condition.

In Figure 3, we showed the escape basins for different values of the parameter *k*; the purple pixels correspond to the initial conditions that escape through the square located in $B_{1}$, the orange one escapes thorough the exit $B_{2}$ and the white ones correspond to initial conditions that do not escape in our computation time $10^{5}$. These points are trapped inside islands. For the case k=0.5 Figure 3a, the basins are separated by invariant curves. In Figure 3b the basins are mixed; However, the initial conditions that are close to the central island tend to belong to $B_{1}$, while the initial conditions close to the points θ=−π and θ=π tend to belong to the $B_{2}$ exit. The intermixing of the basins increases with the increment of *k*, as shown in Figure 3c,d. Figure 4 shows the fractal structure form by the basins in a fine scale.

The escape basins are mixed at arbitrarily fine scales, as is also the escape time, i.e., the number of map iterations that an orbit takes to reach one of the openings has a complicated distribution in the phase space. The time that an initial condition takes to leave the system is shown in Figure 5 (in a color bar), as a function of the initial condition. Reddish colors correspond to higher escape times. This occurs around the islands and the invariant curves in Figure 5a. Bluish colors correspond to a small escape time, and white pixels are orbits that do not escape. It is clear the formation of escape channels, paths to each of the initial conditions, escapes for very small times.

## 4. Basin Entropy

In order to quantify the final state uncertainty produced by the fractality, we apply the concept of basin entropy, developed by Daza and coworkers [20,21]. It was originally developed for basins of attraction and their boundaries, but it can be extended for a more general setting, which is basins of behavior. For open systems, for which the desired behavior is the escape of orbits, we can work with the corresponding escape basins and their boundaries. We have applied this methods in a variety of problems involving magnetic field lines in Tokamaks [34], drift motion of charged plasma particles [35] and light scattering through black holes [36]. Moreover, the basin entropy serves as a means to classify basins of escape (or attraction, in dissipative systems), using the fact that each type of basin maximizes one aspect of the basin entropy [37]. The classification provides a framework for understanding the unpredictability associated with different types of basins, and to deepen our understanding of concepts such as fractality and smoothness, Wada boundaries [38], riddled basins [39] and more [40].

Let us consider a bounded phase space region A, which includes a part of the escape basin boundary, and cover this region into boxes by using a mesh of M×M points. We assign to each mesh point a random variable, whose values characterize each different escape. The basin entropy is obtained by applying the information entropy definition to this set. For open systems, we consider a number $N_{A}$ of exits through which orbits can escape. We assign to each mesh point (which stands for an initial condition) an integer (pseudo-)random variable (called *color*) labeled from 1 to $N_{A}$.

Region A is covered with a regular grid of *N* boxes with sidelength ε=n/M, where n∈N. Let $p_{i,j}$ denote the probability that the *j*th color is assigned to the *i*th box, where i=1,2,…. The fraction of points $p_{ij}$ belonging to a basin inside a box *i* is computed for each box, considering that the colors inside a box are equiprobable, i.e., there is statistical independence. The information (Gibbs) entropy of the *i*th box is
(14)$$S_{i}=-\sum_{j=1}^{m_{i}}p_{ij}\;\log p_{ij}.$$
where $m_{i}\in \left[1,N_{A}\right]$ is the number of colors for the *i*th box. The total entropy for the mesh covering the region A results from summing over the *N* boxes, or $S=\sum_{i=1}^{N}S_{i}$. The basin entropy is defined as the total entropy divided by the number of boxes
(15)$$S_{b}=\frac{S}{N}=-\frac{1}{N}\sum_{i=1}^{N}\sum_{j=1}^{m_{i}}p_{ij}\;\log p_{ij}.$$

The system considered in the present work has two exits, named $B_{1}$ and $B_{2}$, with the corresponding escape basins, as described in the previous section. The corresponding probabilities $p_{i,j}$ satisfy $p_{i,1}+p_{i,2}=1$ for each *i*, such that the basin entropy reads
(16)$$\begin{array}{cc} S_{b} & =-\frac{1}{N}\sum_{i=1}^{N}\left(p_{i,1}\log\left(p_{i,1}\right)+p_{i,2}\log\left(p_{i,2}\right)\right),\\ \; & =-\frac{1}{N}\sum_{i=1}^{N}\left\{p_{i,1}\left[\log\left(p_{i,1}\right)-\log\left(1-p_{i,1}\right)\right]+\log\left(1-p_{i,1}\right)\right\}=-\frac{1}{N}\sum_{i=1}^{N}S_{i}. \end{array}$$

From the computational point of view, the escape basins are discretized into pixels with equal size, such that each square box contains $N_{p}^{2}$ pixels, where $N_{p}$ is the number of pixels contained by the box with sidelength ϵ. For any given box *i*, the corresponding probability $p_{i,1}$ takes on a discrete value out of the following set
$$p_{i,1}\in \left[0,\frac{1}{N_{p}^{2}},\frac{2}{N_{p}^{2}},\cdots ,1-\frac{1}{N_{p}^{2}},1\right].$$ Notice that those boxes for which $p_{i,1}=0$ or 1 do not contribute to the computation of the basin entropy $S_{b}$ because $S_{i}=0$ for such cases. What remains is the contribution of the boxes at the escape basin boundary, namely those containing pixels of both escape basins. Hence, the possible values of the probabilities $p_{i,1}$ for the remaining $N_{b}$ boxes are given by
(17)$$p_{m}=\frac{m}{N_{p}^{2}},\;\left(m=1,2,\ldots N_{p}^{2}-1\right).$$

Considering that a given fraction $q_{m}$ of the $N_{b}$ boxes has a basin probability given by (Equation 17), the basin entropy (Equation 16) becomes
(18)$$\begin{array}{cc} S_{b} & =-\frac{1}{N}\sum_{m=1}^{N_{p}^{2}-1}q_{m}\;N_{b}\;S_{m}\\ \; & =-\frac{N_{b}}{N}\sum_{m=1}^{N_{p}^{2}-1}q_{m}\;S_{m}=-C\;\frac{N_{b}}{N}, \end{array}$$
where *C* is a constant that depends on the distribution of the quantity $q_{m}$. For fractal basin boundaries, which is just the case of the escape basins investigated here, the values of $q_{m}$ are concentrated around a mean value with a Gaussian-like distribution.

Let us denote by *d* and *D* the box-counting dimensions [41] of the escape basin and its corresponding basin boundary, respectively. Considering that it takes a number *N* of boxes with sidelength ϵ in the phase space region A, it follows that $N∼\tilde{N}\epsilon^{-d}$ for small enough ϵ, where $\tilde{N}$ is a constant. By the same token, since it takes a number $N_{b}$ of those boxes to cover the corresponding basin boundary, then $N_{b}∼{\tilde{N}}_{b}\epsilon^{-D}$, where ${\tilde{N}}_{b}$ is another constant and ϵ is also small enough. Substituting both expressions into Equation (Equation 18), we obtain a relation between the basin entropy and the box-counting dimensions of the basin and its corresponding boundary.

This equation can be further transformed by using the concept of uncertainty exponent. Since each initial condition is determined, in the two-dimensional phase space, up to a given uncertainty ϵ, it can be represented by a disk of radius ϵ centered at that initial condition. If this disk does not intercept the basin boundary, all its interior points converge to the same escape and the initial condition is ϵ-certain. Otherwise, if the disk intercepts the basin boundary, it is called ϵ-uncertain. The fraction of ϵ-uncertain disks is known to scale with the uncertainty δ as a power-law: $f\left(\epsilon \right)∼\epsilon^{\alpha }$, where α=d−D is called the uncertainty exponent. Substituting into (Equation 20) there results
(19)$$\ln S_{b}\left(\epsilon \right)=\alpha \ln\epsilon +\ln\left(\frac{{\tilde{N}}_{b}}{\tilde{N}}C\right).$$

We used linear relation Equation (Equation 19) to estimate the uncertainty exponent α for the escape basins of the Weiss map considered in Section 2. For each value of the box sidelength ϵ, we computed the basin entropy using (Equation 16), and we repeated the procedure for a number of values of ϵ with M=1000 and *n* in the interval [15,35], the results being depicted in Figure 6. We have used a least squares fit to obtain a value of α=0.0066±0.0003 in (a) and α=0.0054±0.0002 in (b), where the numerical error arises from the fitting.

The fractality is quantified with the aid of the basin entropy. We consider a grid of boxes, so that each box contains 25 initial conditions. From Daza et al. [20], this value produces the optimum results of the basin entropy. In Figure 7a, we showed the entropy as a function of the parameter *k*. The basin entropy is zero until *k* reaches a critical value $k_{c}$ where the shearless curve is broken, leading to the mix of the two basins. We estimate the value of $k_{c}$ as the value of *k* that produced $S_{b}\neq 0$, meaning that the basins are mixed. This value is $k_{c}=0.535$. The entropy sets close to the maximum value ln2, meaning that there is a great uncertainty of the final state, caused by the fractal structures of the system. In Figure 7b, a magnification of the basins for $k_{c}$ is presented, in the region where the invariant curves existed, but now that is a mixture of the two basins.
(20)$$S_{b}=-C\;\frac{{\tilde{N}}_{b}}{\tilde{N}}\;\epsilon^{d-D}.$$

In Figure 7a, the average escape time $\bar{t}$ as a function of the perturbation parameter *k* is also shown, as well as the basin entropy $S_{b}$ in function of the same parameter. While the entropy goes from zero to almost the maximum value, the mean escape time has an extreme in *k*∼0.4. This is most probably caused by the stickiness effect around the invariant curves and islands in the phase space that trap the orbits for long periods of time. After the last invariant curve is broken, the basin entropy approaches the maximum theoretical value ln2 and the mean escape time increases to a higher value. The entropy close to its maximum means that the final state unpredictability of the system is very high and that the boundary of the basins is an area filling curve with an almost zero uncertainty exponent. Moreover, the large mean escape time suggests suggests that the trajectories are very close to the stable manifold of the chaotic saddle, given that this is the closure of the basins boundary area filling curve, high entropy implies in high mean escape time.

The low uncertainty exponent and the high entropy indicates that the opening causes the system to practically become nondeterministic. This is an effect of the size of the exits. Aguirre and Sanjuán [33] found that the unpredictability grows indefinitely as the size of the exits decreases and tends to zero. This leads to total indeterminism, which is a general feature of open Hamiltonian systems.

## 5. Conclusions

In this work, we investigated the escape of chaotic orbits in a non-twist map called a Weiss map. Considering the opening in the phase space, we can calculate the escape basins. They present a fractal structures given by the underlying dynamical structure of the chaotic orbits. The escape time also showed a fractal structures with the presence of paths where the escape is very fast.

In order to quantify the fractality, we used the concept of basin entropy, a quantity of the uncertainty of the final state caused by the fractal structures. Moreover, we showed a way to compute the uncertainty exponent with the basin entropy. For k=0.6, α=0.0066, which indicates that the basin boundary is extremely involved. The system exhibits a collection of invariant curves that act as boundaries between chaotic regions surrounding two main islands. However, as *k* increases, these curves become broken, until only one curve remains the shearless curve. This is broken when $k=k_{c}$, so that the two chaotic regions are connected. The values of $k_{c}$ were estimated using the entropy basin concept, to which we found the value of $k_{c}=0.53$.

The basin entropy concept we used in this work is based on the Boltzmann–Gibbs –Shannon–von Neumann version of the entropy, which is an extensive quantity by definition. However, in dynamical systems exhibiting complex behavior, including coexistence among periodic, quasiperiodic and chaotic orbits such as the Weiss map we considered in the present paper, it has been argued that a non-extensive entropy would be better suited, such as Tsallis entropy [42,43]. This is particularly interesting when the chaotic transport is characterized by anomalous diffusion, which is heavily influenced by stickiness and other dynamical features of the chaotic orbit [44]. A further extension of the present approach would be, therefore, to adapt the basin entropy concept *vis-a-vis* of the Tsallis non-extensive entropy.

## Figures and Tables

**Figure 1 entropy-25-01142-f001:**
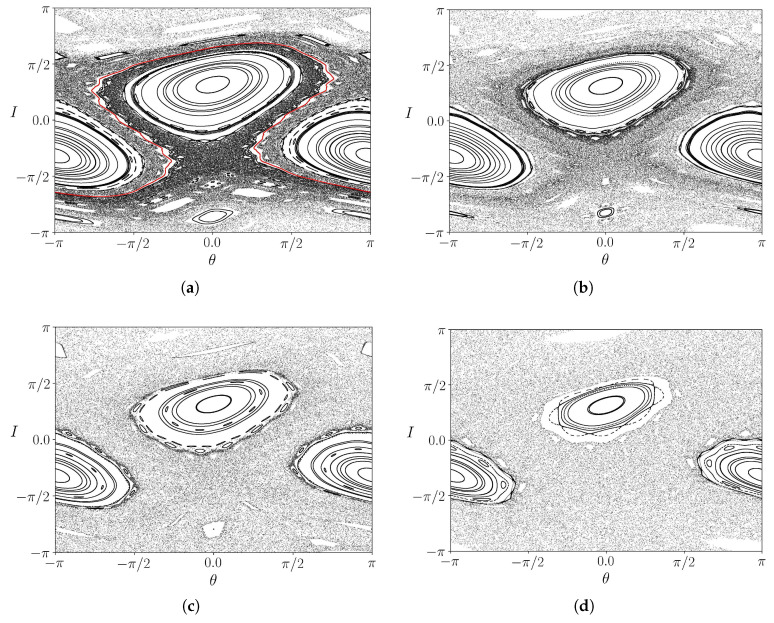
Phase space of the map for (**a**) k=0.50, (**b**) k=0.55, (**c**) k=0.60 and (**d**) k=0.70. The red line in (**a**) represents the shearless curve, which separates the two chaotic regions.

**Figure 2 entropy-25-01142-f002:**
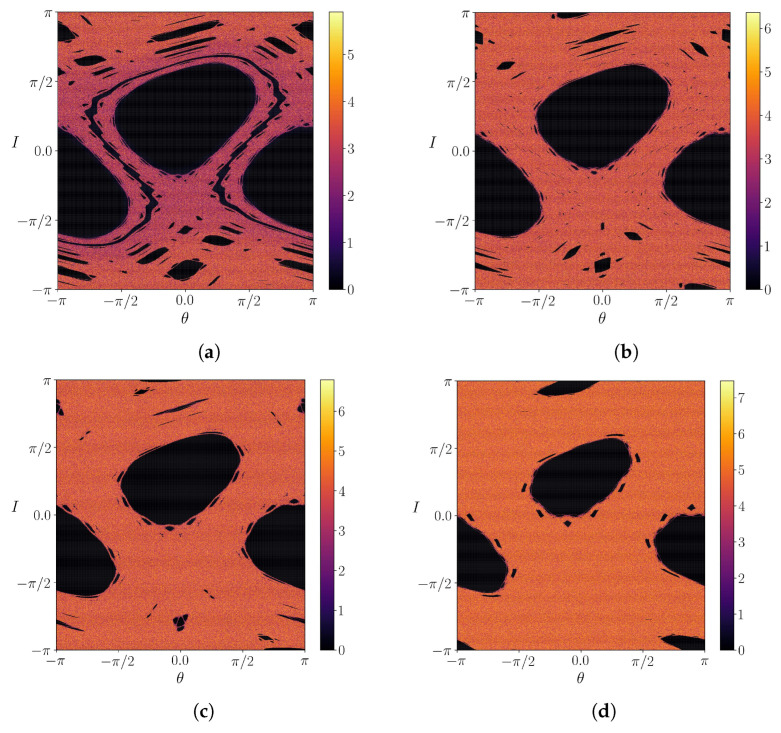
Largest Lyapunov exponent for (**a**) k=0.50, (**b**) k=0.55, (**c**) k=0.60 and (**d**) k=0.70.

**Figure 3 entropy-25-01142-f003:**
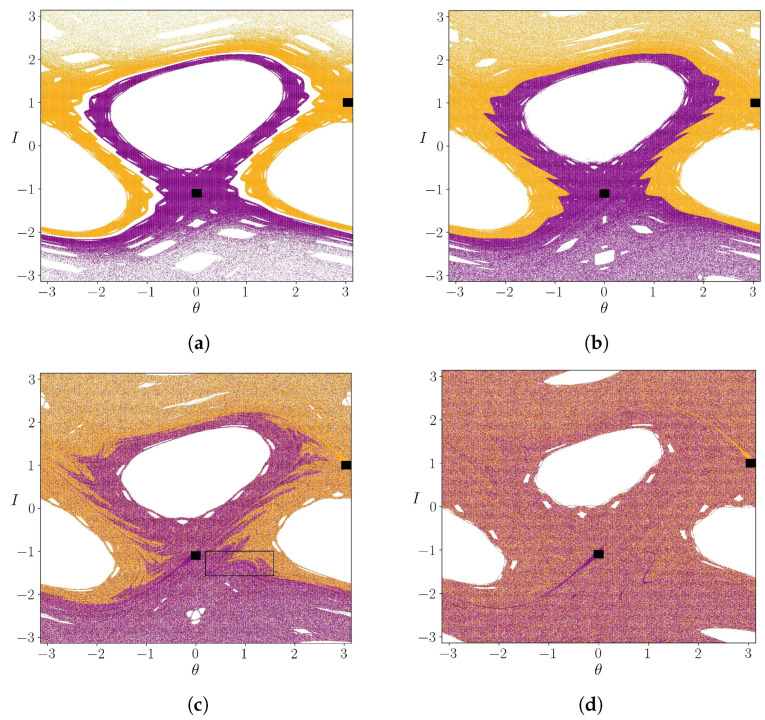
Escape basins for k= (**a**) 0.50, (**b**) 0.55, (**c**) 0.60 and (**d**) 0.70. Purple pixels escape through $B_{1}$ the internal region, close to the central island, orange pixels belongs to $B_{2}$. White pixels are points that do not escape, because they are inside islands. Black squares represent the exits and the the black frame in (**c**) is show in detail in Figure 4.

**Figure 4 entropy-25-01142-f004:**
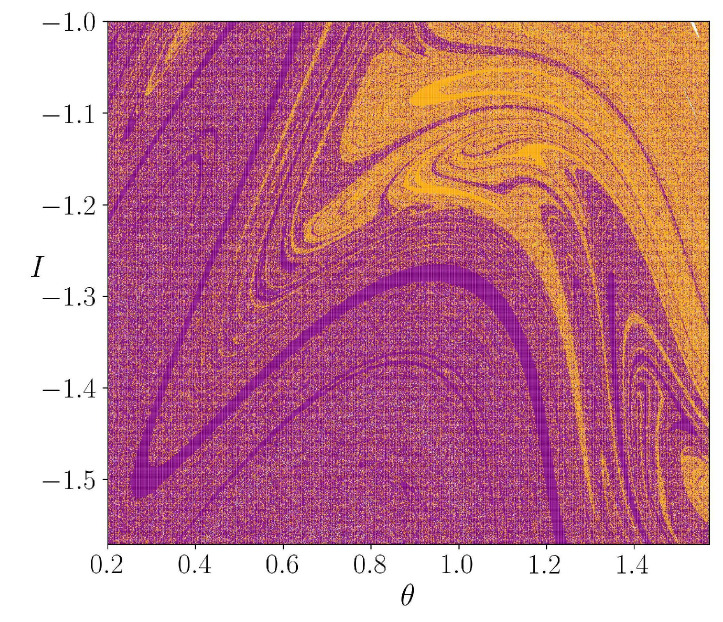
Zoom of the rectangular in Figure 3c. The purple and orange basins are intermixed at a fine scale, with a fractal pattern.

**Figure 5 entropy-25-01142-f005:**
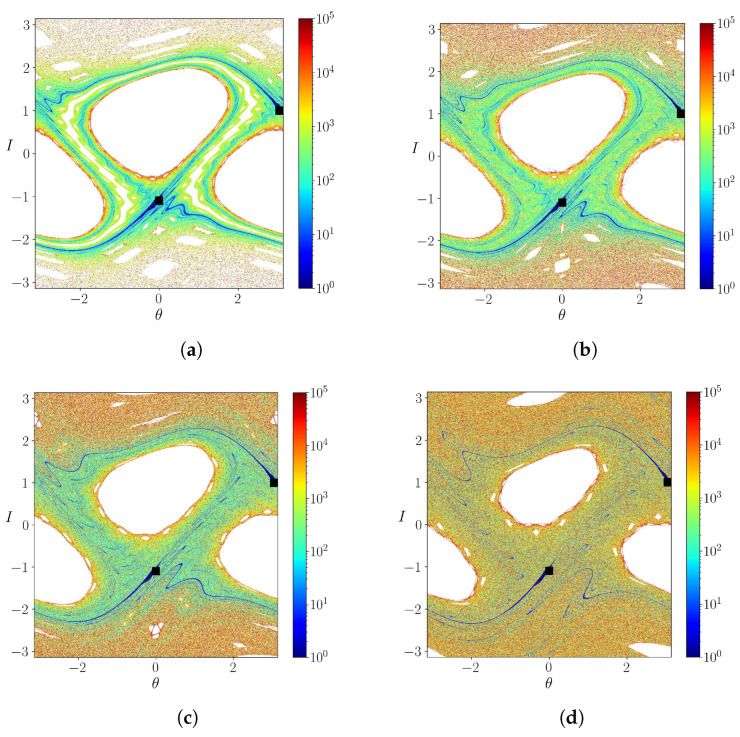
Time to escape for k= (**a**) 0.50, (**b**) 0.55, (**c**) 0.60 and (**d**) 0.70. The color bar indicates the number of iterations of the map until an initial condition reaches one of the openings. Red colors correspond to a high number of iterations and blue colors to a small number. Black squares represent the exits.

**Figure 6 entropy-25-01142-f006:**
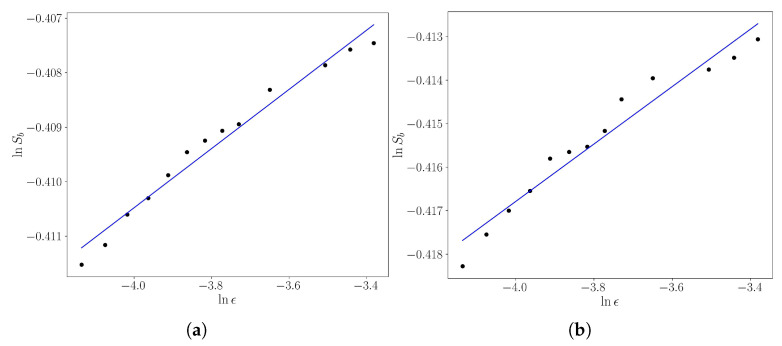
Basin entropy as a function of the sidelength ϵ for the Weiss’ map with (**a**) k=0.6 and (**b**) k−0.7. The blue line is a least squares fit.

**Figure 7 entropy-25-01142-f007:**
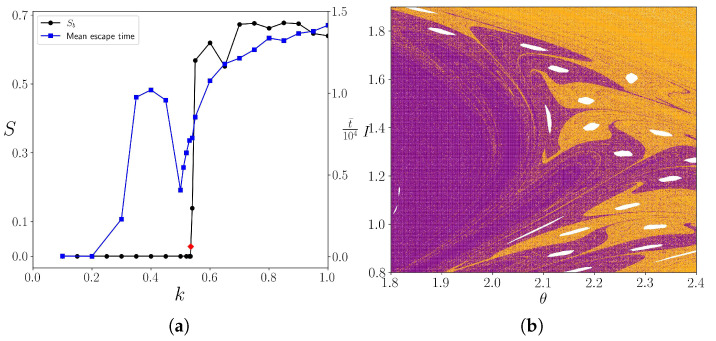
(**a**) In black: basin entropy as a function of the parameter *k* for boxes containing 25 initial conditions each. The red diamond is the first non-zero value where the shearless curve is broken, corresponding to k=0.535. In blue, the mean escape time as a function of *k*. (**b**) Zoom-in of the basins for k=0.535, showing that there is a mixture of the two basins.

## Data Availability

The data used in this work are available from the author, L.C.S., upon reasonable request.

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
