# Peer review of "Basin Entropy and Shearless Barrier Breakup in Open Non-Twist Hamiltonian Systems"

_entropy, 2023, doi:10.3390/e25081142_

Round 1

Reviewer 1 Report

The article presents the study of a non-twist map using the basin entropy. Given the conservative nature of the map, the authors open the system in order to create to distinct basins and compute the entropy associated as parameters are changed. The article needs profound changes before its publication can be considered, here are my main comments:

- The use of the basin entropy in this context feels somewhat artificial. In particular, the authors fail to explain why they open the system in that specific way. They simply indicate that basins can be smooth or fractal depending on the periodic or chaotic nature of the trajectories in the selected region of phase space. Would the results be different for a different choice of the exits? What about three exits? What about bigger/smaller exits? What is the physical meaning of each case?

- The exit time and the basin entropy are both computed. However, not much is discussed about its relation. It would be nice to have a quantitative comparison of both magnitudes.

- Authors find basins with uncertainty exponent almost equal to zero and nearly maximum basin entropy. In principle, this is expected in this kind of artificially opened systems as the exits shrink, as explained in "Aguirre, J., & Sanjuán, M. A. F. (2003). Limit of small exits in open Hamiltonian systems. Physical Review E, 67(5), 056201". Also, it could be interesting to contextualize the results with respect to the classification scheme proposed in "Daza, A., Wagemakers, A., & Sanjuán, M. A. F. (2022). Classifying basins of attraction using the basin entropy. Chaos, Solitons & Fractals, 159, 112112."

In my opinion, the authors do not sufficiently motivate the study of the system nor the use of the basin entropy. All results are expected and their physical consequences are barely explored. Therefore I cannot recommend the publication of the manuscript in its present form.

The paper is in general well written, although there are some spelling mistakes scattered along the manuscript that require careful revision. In particular, the last paragraph of the results section (just before the conclusions) contains many errors, such as:

- "is quantify" should be "is quantified"

- "k reach a critical" must be "k reaches a critical"

- "this values" needs to be replaced by "this value"

Round 2

Reviewer 1 Report

In the revised version, the authors have adequately addressed most of the points raised in my previous review, including a short discussion about the choice of the exits and their size. Also, they  briefly mentioned some of the information that can be extracted from the basin entropy calculations. The new data included in Fig. 7a shows a transition at kc for both the escape time and the basin entropy, indicating the change in the dynamics.

However, I think that there is an error with respect to the expected behavior of the basin entropy as the exit size varies. In the text, it is said that "smaller widths would result in slightly lower values of the basin entropy". I believe that it would lead to slightly higher values of the basin entropy, and their comment in the cover letter seems to support this point.

Although the authors have ammended some of the small errors and typos of the first version, the new paragraphs contain many mistakes and should be carefully revised. Here I will just list a few examples:

- Line 186 and others (251, 255...): "the basins entropy" should always be "the basin entropy".

- Line 188: "the classification provide" must be "the classification provides".

- Line 252: "the mean escape time have a extreme" must be "the mean escape time has an extreme".

- Line 258: "a area filling curve with a almost zero" should be changed to "an area filling curve with an almost zero".

- Line 259: "the high great mean escape time suggest" would be better as "the large mean escape time suggests".

Reviewer 2 Report

The revised version takes into account the suggestions of the referee reports. The quality of presentation has been significantly improved and I suggest acceptance.